# OpenReview forum: "Log-Polar Space Convolution"
_ICLR.cc/2022/Conference — ICLR 2022 Submitted_

### Official Review · Reviewer_2ZGD · 2021-11-01

**Correctness:** 3
**Technical Novelty And Significance:** 3
**Empirical Novelty And Significance:** 2
**Recommendation:** 5
**Confidence:** 4

**Main Review:**

Positives
---------

Overall this is a nice idea, and the experimental results are fairly convincing in showing that the proposed method does not hurt in existing model architectures. I think that this provides a nice new potential "tool in the toolbox" for future investigations.


Concerns
--------

Broadly speaking my concerns fall into three areas:

- __Writing/Presentation__: The mathematical parts of the paper need revisiting/correcting; in the current version equations 1 and 3 need to either incorporate a dot symbol, a transpose on the X, or an extra summation over channels to make sense. Additionally, from a mathematical perspective the equations are those of cross-correlation rather than convolution, so it would probably be better to either use the cross-correlation operator symbol or reverse the indexing of the kernel (or input). Some mention of this in the text would be wise, and it may be better to for example change the title of the paper to something like "log-polar space convolutional layers" to make it absolutely clear that the paper is about CNN layers rather than more general convolutions. Further, the authors need to clarify the use of the bias term in their implementation - it is not clear from the math/text if there is one or not.

- __Results__: Overall the results come across as weakly positive; The results on imagenet and voc2012 feel rather superficial without quantification of variance given the gains are so marginal. I realise that this might exceed the computational resources available to the authors to acheive, however the marginal gains are deserving of a much more detailed discussion. In particular, I'd like significantly more discussion and analysis of what the numerical accuracy/auc/etc metrics mean in light of the what is shown in figure 4, where it appears that a significant number of the kernels do not benefit from the increased receptive field size. Also regarding fig 4, it is not clear quite how this was created - does a mid-grey value represent kernel values of zero, or are the values somehow normalised? This is important as it would completly change one's interpretation of the results.

- __Complexity Analysis__: Firstly, I'm not sure I understand the reported FLOPs result - why is the dilated convolution much higher? The number of FLOPs to apply a standard convolution and a dilated convolution with the same number of weights should be identical. Further, I suspect that this analysis would only tell part of the story about complexity. More specifically one would want to understand the actual latency of the different approaches because the memory access patterns are going to be significantly different. In terms of wall time, how do the different methods compare during training and inference for example?


**Summary Of The Paper:**

The paper proposes a log-polar space convolution in which elliptical kernels are convolved over the input. The kernels are broken into radial segments and each value of each segment is applied to all the pixels in the underlying portion of the image, which are then summed and potentially normalised. Actual implementation is achieved through use of log-polar space pooling which can be used to create a set of pooled featuremaps to which a regular convolution can be applied. The advantage of such an approach is that it allows for much larger receptive fields without the need forr more weights.


**Summary Of The Review:**

Overall, I think there are some interesting ideas emerging from this work, but in its present state it's a bit borderline. As outlined above if the problems with presentation could be addressed, and further information given on the computational complexity with more analysis/discussion of the results then my concerns would likely be assuaged.

---

### Official Review · Reviewer_pwSX · 2021-11-02

**Correctness:** 3
**Technical Novelty And Significance:** 3
**Empirical Novelty And Significance:** 3
**Recommendation:** 5
**Confidence:** 4

**Main Review:**

Strength
1. The idea of LPSC is interesting and seems reasonable to me. It does provide a different perspective of performing convolutional operators, while the ability to exponentially expand local receptive field (LRF) without significantly increasing the parameters can be potential in image recognition tasks, especially for dense prediction ones like image segmentation.
2. The performance of LPSC-based networks has surpassed that of original ones on a variety of architectures and on different datasets of classification/segmentation tasks, which can be a good sign of generality.
3. The idea is straightforward to understand and the paper writing is generally easy to follow.

Weakness
1. The ability to expand LRF is a major claim, however, the current experimental results may be insufficient to support the claim. The authors only evaluate their methods on light-weight network structures like ResNet-18/20, MobileNet, etc., while the receptive fields in those models might not be dense or broad enough and therefore can be compensated by the proposed LPSC. However, what about those more complicated models like ResNet-101 that may already have a good LRF coverage? Will LPSC still work on them?
2. Expanding LRF is a popular topic in semantic segmentation, and there are some missing related works like Dense ASPP [a] or [b].
3. The computational inefficiency can also be a concern here. The authors have already discussed that the spatial complexity can be a limitation (which I appreciate), but I think this also indicates an increased time complexity since the mask has to be pre-computed. Besides, as the radius grows, the area of outer rings can grow exponentially, which can lead to an exponentially-increased complexity.
4. The log-polar form of LPSC kernel assumes that the features near the centre can be more important than those more distant ones, since the learnable parameters are much dense in central regions. However, I am not sure why this kind of data distribution assumption can work better than the normal convolutional operators or other operators like deformable or dilated convs. I think a discussion on the data distribution can be necessary.
5. The achieved improvement on some datasets is somehow marginal. The lack of more advanced models like ResNet-101 or Xception-65 can make the results unconvincing. The authors only compared their method with dilated convs in segmentation experiments, but I think it can be important to also compare with other related methods like DenseASPP [a], Deformable Conv or Self-Attention module.

[a] Yang, Maoke, et al. "Denseaspp for semantic segmentation in street scenes." Proceedings of the IEEE conference on computer vision and pattern recognition. 2018.
[b] Chen, Liang-Chieh, et al. "Searching for efficient multi-scale architectures for dense image prediction." arXiv preprint arXiv:1809.04184 (2018).

**Summary Of The Paper:**

This paper proposes a Log-Polar Space Convolution (LPSC) for general image recognition tasks. The proposed convolutional kernel distributes in the form of log-polar spaces, and therefore the receptive field can expand exponentially as the radius grows without significantly increasing the learning parameters. To implement LPSC, the authors first utilize a log-polar space pooling to pre-compute an up-sampled feature map and then LPSC is performed just like normal convolutional filters.  The proposed LPSC are integrated into several popular CNN backbones like ResNet, VGGNet, MobileNet, etc., and the performance of modified networks has exceeded the original structure on several image recognition and segmentation datasets.

**Summary Of The Review:**

Overall, the idea of this paper is interesting and reasonable, while the proposed LPSC can be potential for dense prediction tasks. However, there are several limitations of this method, while the experimental results are not very convincing at this stage. For now, I will vote for a marginally below.

---

### Official Review · Reviewer_TRTh · 2021-11-02

**Correctness:** 3
**Technical Novelty And Significance:** 3
**Empirical Novelty And Significance:** 2
**Recommendation:** 5
**Confidence:** 4

**Main Review:**

The main idea of this paper is sound and intuitive, and the empirical results on multiple models and datasets demonstrate the superior performance of the proposed paper. However, some important aspects are not carefully considered or explained in the paper:

1. The authors claim that they use the same meta-parameters for all methods for a fair comparison. However, this claim is problematic. It is well known that the meta-parameters can significantly affect the performance of machine learning models, and proper tuning of meta-parameters is a necessary step in practice. Using the very same meta-parameter may lead to non-informative results, because no practical model will be trained this way and different models will require different meta-parameters. Unless the authors can show that all the methods are affected equally by the meta-parameters, which is rarely the case, the evaluations provide very limited information. Also, the authors seem to report the best result in ablation study as the final performance, which optimizes the model performance on the test set and may lead to overly optimistic estimation for the performance.

2. Comparing models with the same number of parameters may not be fair. Note that the proposed method can be considered as a special case of separable convolution with a fixed, carefully initialized weight for the depth-wise convolution. Therefore, the effective capacity of the model is larger than a model with plain 2D convolution. The authors should discuss how these fixed convolution layers (i.e. log-polar space pooling) affect the performance, e.g. is this the best weight, how does learned weights compare with fixed weights. Also, the authors should also consider other aspects of the model such as the computational cost, memory usage, etc. to provide a more complete picture for the pros and cons of the proposed method.

3. The authors only evaluate the performance on relatively shallow models, i.e. ResNet-20. Because deeper models may learn the effective receptive field better, the proposed method may have diminished gain as the model becomes deeper or even harm the performance. Also, more recent architectures such as transformer and active convolution may be able to learn the appropriate receptive field directly. Given that the trend of architecture design is to use deeper and / or more flexible architecture and learn the regularization from data, the authors should discuss the implication of the proposed method on more recent models.

4. Related to 1 and 3, the performance of the baseline methods seem to be sub-optimal. Therefore, the performance gain might not be very meaningful as the gap may reduce as the baseline models are properly trained.


**Summary Of The Paper:**

This paper introduces a new formulation for the convolution operator. Instead of defining convolution kernels on a regular 2D grid, this paper proposes to define convolution kernels on a circle divided by angle and radius. Note that the radius is divided equally in log-scale to account for the observation that content near the center should have more impact on the outputs. The new convolution operator is generic and can replace existing 2D convolution operators. Also, it can be implemented in existing frameworks easily using a customized pooling operator plus standard convolution. Empirical results on 1) multiple architectures, 2) multiple tasks, and 3) multiple datasets show that the proposed method outperforms standard convolution under a similar number of parameters.

**Summary Of The Review:**

The method introduced in this paper is intuitive and seems promising empirically. However, some important aspects are not carefully considered in the evaluation. Therefore, it is hard to justify the benefit and contribution of the proposed method.

---

### Official Review · Reviewer_MHsy · 2021-11-06

**Correctness:** 3
**Technical Novelty And Significance:** 2
**Empirical Novelty And Significance:** 2
**Recommendation:** 3
**Confidence:** 4

**Main Review:**

The paper is clearly written, and the main idea seems like a logical thing to try. There are a variety of precedents for log-polar features such as the organization of the human retina and the success of classical methods like shape contexts. The authors are honest about the shortcomings (such as large memory requirements) of their method.

While there is nothing terribly objectionable about this paper, I'm afraid the result is not terribly compelling. While the exact architectures proposed by the authors are not directly implementable with standard techniques in the literature, one can come awfully close.

For example, if one makes an architecture with 3 types of dilation (say, dilations of 1, 2, and 4) in a particular layer, and makes all of the filters relatively small, then the next layer could assemble the outputs of such filters to learn something extremely similar to the filters prescribed in this paper. Now, a counterargument might be that in such a dilation-based architecture, the network might not learn to put together filters in a way that mimics the log-polar structure produced here. On the other hand, if such an architecture does not learn to assemble filters in this way, the counter-counterargument would be that these filters are not the most effective filters to reduce the loss function.

Irrespective of this argument, I would expect the authors to provide stronger baselines. For example, a strong baseline would be, "how could I build an architecture, using traditional methods, that would allow filters to be learned that incorporated dense sample around a point combined with sparse sampling far from the point?" Such a strong baseline would be more interesting that simple dilation architectures.

The main point of all of this is that it seems highly unlikely that such a specific structural change to an architecture would allow a network to learn functions that another standard network could not learn. While the experiments support the idea that the presented architecture has a minor advantage over other simple baselines, it by no means makes a compelling argument that such architectures are interesting from a real practical point of view on the tough problems people are currently addressing. In short, the argument that this is a compelling new architectural element has not been made.

And if the network is not more expressive, then the question becomes whether it is significantly faster to train or does it require less memory? Apparently, this network offer none of these advantages and even some disadvantages. As such, it seems unlikely that anyone would adapt it for a specific task over current state of the art models.



**Summary Of The Paper:**

This paper revisits the old of idea encodings that have higher density around a particular point and lower density as one moves away from a particular analysis point. In the literature, these have often been called log-polar filters, and including classical computer vision techniques such as shape context features. The authors bring these types of filters to CNNs, by modifying architectures (using pooling techniques) to allow for filters that have this type of structure.  They do a variety of experiments showing that these methods can outperform certain simple baselines, such as local filters and dilated filters.


**Summary Of The Review:**

The paper is well-written with a somewhat interesting idea, but ultimately, it seems unlikely that such an architecture provides advantages over strong baselines, which are not really explored.

---

### Decision · Program_Chairs · 2022-01-20

**Decision:**

Reject

**Comment:**

This paper proposes an alternative for constructing convolution kernels: instead of uniform spatial resolution, it proposes a spatially varying resolution with higher precision at the center of the kernel. The resolution decreases logarithmically as a function of the distance to the center. All reviewers agree that the idea is interesting, but in its current form, the submission is not mature enough to be published.

In particular, reviewers raised some concerns about computational efficiency of the method. The authors explain that their method runs slower than conventional convolution because the implementation uses of-the-shell conventional convolution modules, and they speculate that the speed can be accelerated if the method is directly implemented with CUDA or by directly adapting the underlying code of convolutions in the integrated framework. While this is a reasonable argument, it is not actually verified. This it is not clear if there would be other road blockers to achieve the promised performance. It would be great if authors could present actual performance of the method using either of their suggested solutions (CUDA or modifying code of convolutions).

In addition, reviewers raised concerns about some aspects of the evaluation setup, where test data is used to report the best performance. Authors respond that baselines are trained in the same fashion, hence the comparison is still fair. However, the reviewers were not convinced by this response. In concordance, I also think the use of test data during training is misleading, even if all methods use the same strategy, because this may tell us more about which approach can better (over)fit to the data as opposed to how well the methods are able to generalize to unseen samples.

Another concern relates to the diminishing return in the performance as networks get larger. The authors respond that this might be because only the first layer uses the proposed log-polar convolution, speculating the problem will go away if the proposed approach is used in all layers. However, this is not empirically verified again and remains unclear if this is indeed the reason.

I suggest authors resubmit after accommodating the provided feedback.